# Delivery of community-centred public mental health interventions in diverse areas in England: a mapping study protocol

Fiona H Duncan ![ORCID],[1] Mike McGrath,[2,3] Cleo Baskin,[4] David Osborn,[2] Jen Dykxhoorn,[2,3] Eileen F S Kaner,[5] Shamini Gnani,[4] Louise LaFortune,[6] Caroline Lee,[6] Kate R Walters ![ORCID],[3] James Kirkbride,[2] Laura Fischer,[7] Oli Jones,[7] Vanessa Pinfold,[7] Jude Stansfield ![ORCID],[8] Emily J Oliver ![ORCID],[1] On behalf of the NIHR SPHR Public Mental Health Programme.

For numbered affiliations see end of article.

**Correspondence to**
Dr Fiona H Duncan;
fiona.h.duncan@durham.ac.uk

## ABSTRACT

**Background** Public mental health (PMH) is a global challenge and a UK priority area for action. However, to progress, practitioners require a stronger evidence base regarding the effectiveness of approaches, particularly regarding promotion and prevention through community-centred interventions. In addition, policy-makers need to understand what is being delivered, particularly in areas of high need, to identify promising practices or gaps in PMH provision. Finally, and importantly, the public need better information regarding what approaches and services are available to them. We report a protocol designed to (1) identify the types of community-centred interventions used in purposively selected diverse geographical areas of England to improve PMH outcomes and (2) describe the type, target population, content and outcome measures of each intervention.

**Methods and analysis** Five local authority areas of England were selected based on either high social deprivation or differing ethnic population statistics and geographical locations. Community-centred interventions in each area will be identified through: (1) desk-based data capture from standardised searches of publicly-available information (eg, policy, strategy and intervention advertising), (2) established professional networks and service contacts, (3) chain-referral sampling of individuals involved in local mental health promotion and prevention and (4) peer researchers, who will use their personal experience and local knowledge to help identify potentially relevant organisations. Data on the key features of the interventions will be extracted from individuals either by structured interviews or by electronic questionnaires with information regarding the intervention(s) of which they have knowledge. Initial data analysis will involve tabulating descriptive information and grouping interventions according to intervention type, target population, risk/protective factor and intended primary outcome. A descriptive comparison will be made between selected geographical areas.

**Ethics and dissemination** Ethical approval was obtained from Durham University's Department of Sport and Exercise Sciences Research Ethics Committee. We plan to

### Strengths and limitations of this study

► Multiple sources are used to collect data.
► The protocol has been piloted in two areas and refined.
► A range of geographical areas of England has been included.
► It cannot be guaranteed that key individuals will be able to provide accurate information or that websites will be up-to-date.
► Only services that explicitly aim to improve the mental health of adults have been included (services aimed at only children and adolescents have been excluded).

disseminate our findings at relevant conferences, meetings and through peer-reviewed journals. We also plan to disseminate to the public and intervention providers through social media and/or newsletters.

## BACKGROUND

Public mental health (PMH) or population mental health is an emerging field and consequently, there is still a lack of consensus regarding its definition and key components.[1] The WHO's[2] PMH approach emphasises the distinction between mental health and mental illness, stating that mental health should be considered to be more than the absence of illness, and therefore conceptualisations of PMH should include mental well-being promotion as well as mental illness prevention. Mental well-being promotion involves encouraging good mental affect, positive feelings such as life satisfaction and happiness, reducing inequalities, building social capital, enhancing the quality of life and enabling optimal psychological and psycho-physiological development throughout the

life course. Mental illness prevention involves reducing the incidence, prevalence and recurrence of mental health problems, as well as reducing the risk factors and the impact of mental illness on the affected person.[2] Consequently, WHO recommended in its mental health action plan several actions that focused on the promotion of positive mental health and emotional well-being as well as the prevention of mental health issues.[3] Consistent with this, Public Health England (PHE), a UK Government agency, also conceptualises PMH as consisting of the promotion of well-being and good mental affect, and the prevention of mental health problems and suicide.[4] A key aspect of PHE's approach to PMH is the recognition that mental health is a complex determinant of overall health and well-being and therefore actions that aim to minimise risk factors and enhance protective factors of mental health should be incorporated into a wide range of public health policies.[4] In line with the viewpoint of WHO and PHE, the conceptualisation of PMH that will be adopted for this protocol includes adverse and positive mental health.[5]

Improving PMH is important as each year, in England, almost a quarter of adults experience at least one mental health problem[6] and 17% experience a sub-threshold mental health problem.[6] As well as the impact on the individual, this creates a substantial economic and social burden for the country as mental health problems have been reported to be associated with increased risk of physical illness[7 8] such as coronary heart disease,[9] increased mortality,[10] higher rates of health risk behaviours such as smoking,[11] reduced life expectancy,[12] work absence,[13] unemployment,[14] homelessness[15] and reduced quality of life.[16] Conversely, good mental well-being has been reported to be associated with reduced mental health problems[17] and suicide,[18] reduced heart disease,[19] reduced mortality and increased life expectancy,[20] improved recovery from physical illness,[21] increased physical activity,[22] reduced absenteeism[23] increased productivity at work, higher income and stronger social relationships.[17]

Despite the high prevalence of mental health problems, there has been chronic underinvestment in mental health prevention across the English National Health Service (NHS) in recent years[24] and provision of interventions and services that support PMH in local areas have been badly affected by austerity measures introduced by the UK government in 2010.[25–27] There is ongoing concern about how to promote PMH in areas experiencing deprivation, in particular.[25–27]

A range of policy documents and publications have highlighted the importance of addressing the modifiable determinants of PMH within communities.[15 27–29] These modifiable potential factors include both acute and chronic determinants such as economic disadvantage,[6 30] debt and financial difficulties,[31–34] unemployment,[14] social isolation and loneliness,[14] intimate partner violence,[35] sedentary lifestyles,[36 37] fuel poverty,[38] food insecurity,[39 40] homelessness[41] and belonging to an ethnic

minority group and other groups that experience stigma/marginalisation.[14]

If causal, such a wide range of inter-related modifiable factors suggests that a multitude of approaches may be needed to improve mental health. However, the current evidence base for the effectiveness and cost-effectiveness of community-centred PMH interventions for adults is limited.[42] Promising examples of PMH interventions include workplace-based stress management interventions,[43] community interventions for loneliness in older adults,[44 45] and welfare benefits and debt advice interventions based on general practitioner practices.[46] While these show promise to improve mental health and well-being outcomes, the precise 'active ingredients' of these interventions need further evidence.[47] Further, in order to progress the provision of PMH services and approaches we need to understand what is currently being delivered in local areas, so that the assets and capabilities available in different communities addressing modifiable determinants of mental health are accurately mapped. Specifically, we need to identify any promising practice, learning from existing (successful and unsuccessful) practice, inconsistencies, variations or gaps in service delivery and what evidence exists concerning the effect they have on meeting population needs. As part of a larger National Institute of Health Research (NIHR) funded programme through the School for Public Health Research (SPHR), the present study protocol aims to:

1. Identify the types of community-centred interventions used in purposively selected diverse geographical areas of England to improve PMH outcomes.
2. Describe the type, target population, risk/protective factor, content, reach, effectiveness, cost-effectiveness and intended outcome measures of each intervention.

## METHODS
### Study design
A mapping approach was undertaken to identify community-centred interventions across five selected areas of England.

### Selected areas of England
The following local authority areas of England have been selected: Redcar and Cleveland (North East), Cambridgeshire and Peterborough (East), Blackburn (North West) and Camden and Islington, Hammersmith and Fulham (Greater London).

The process of selecting these areas was as follows: (1) a long list of candidate local authority areas in the North East, North West, London and East of England was created to ensure that a wide geographic spread of areas were represented. (2) A range of statistics from the Office of National Statistics annual population survey regarding these local authority areas was examined, including statistics on deprivation. (3) The final areas were chosen as they have diverse social deprivation and/or ethnic populations, were proximal to research group institutions

associated with the NIHR SPHR and therefore, research group members could use established contacts with relevant local authority personnel to maximise response rates.

This protocol was piloted by mapping two of these selected areas (Redcar and Cleveland, Camden and Islington) between August 2019 and January 2020. We plan to have completed the mapping of the remaining areas by the end of May 2020.

### Inclusion and exclusion criteria

The following inclusion criteria will be applied to identify potential PMH interventions:

▶ Any intervention which explicitly aims to improve adult mental health and/or well-being. Such aims may be expressed as: to reduce social isolation, to improve emotional well-being, to increase happiness, to improve resilience, to decrease stress or to improve confidence and/or self-esteem. This aim does not have to be stated in materials that participants have access to for the intervention to be included, it could be stated verbally by an intervention provider. The intervention might consist of support, individual advice giving, signposting, activity groups, social groups or referral to community services. These programmes and interventions can be aimed at the general population or targeted at a specific subgroup that is, older people, particular ethnic minorities, people who have job insecurity/debt/housing problems. Interventions delivered by individuals within the community (eg, workshops and support groups) will be included.

▶ Any intervention which is delivered in, or available to residents of, the target local authority area, even if it is funded, delivered or managed within another local authority area or larger geographical area. Online and digital interventions will be included.

▶ Any intervention delivered by the government, the local authority, third sector organisations, social enterprises, the NHS (if non-clinical) or located in primary care settings (if non-clinical).

▶ Programmes and interventions that are available to local residents during the data collection period.

An intervention will be excluded if, it is:

▶ A clinical mental health intervention or an intervention that is primarily aimed at people identified with a diagnosed or suspected mental health condition.

▶ A wholly (for-profit) private sector-funded intervention.

▶ An intervention that is exclusively aimed at children or adolescents, as this mapping exercise is focused on services for adults.

▶ Improving mental health and/or well-being is not one of the intervention's explicit aims (whether the aim is stated in writing or verbally by an intervention provider).

### Data collection

Two types of information will be sought during the mapping exercise: (1) identification of relevant interventions; (2) data on the key features of the identified services.

(1) Identification of interventions. Much information about local services is unpublished and available information online can rapidly become out of date. Therefore, we will use a range of approaches to create a list of interventions as exhaustive as possible, bringing together information from four sources: (a) first of all, desk-based internet searches of available information resources will be carried out. Local authority, relevant third sector, NHS and Clinical Commissioning Group websites will be searched for key policy documents, strategies, intervention advertising or guides to local services (table 1). These sources will then be scrutinised for evidence of the existence of potentially relevant interventions. The following key search terms will then be entered into an internet search engine: public mental health interventions, community development, community capacity building, community activities, anxiety/depression/stress courses/groups/workshops, well-being workshops/courses/groups/activities/ sessions, life skills courses/classes, mental health promotion, mental health prevention and suicide prevention. Social media platforms will also be searched using these terms. Where available, the contact details of potentially relevant individuals and/or organisations were identified through these searches of available information resources and these general internet searches will be noted and followed-up by email or by telephone for further information about interventions or services that they deliver. (b) Directly approaching by email and telephone potential key informants in each area and requesting information on interventions that fit our inclusion criteria (suggested key informants are presented in table 1). Potential informants include relevant contacts from the public health department of local authorities, Clinical Commissioning Groups, voluntary sector provider organisations and signposting services (eg, link workers or community navigators). The contact details of these informants will be obtained from the relevant organisation website or via previously established relationships between the research team and the organisation. (c) Chain-referral sampling (Snowballing). When individuals involved in local PMH promotion are contacted and respond with information for the study, then they will be asked if they could provide details of any other public mental health interventions or relevant organisations that they are aware of in the specified local authority area. (d) Peer researchers working within the mental health programme of the School for Public Health Research will use their personal experience of managing poor mental health and their local knowledge to assist in the identification of potentially relevant organisations or individuals. They will also contact these organisations to obtain information about services and follow-up with services already identified to obtain further information.

(2) Data on the key features of the interventions. After identifying eligible interventions, we will extract information from websites, local authority documents and key

**Table 1** Key policy, strategy and signposting documents from each selected area to be searched and examples of potential key local authority departments and organisations to be approached

| Key documents to be searched online | Key local authority departments to be contacted | Key third sector organisations to be contacted |
|---|---|---|
| ► Public mental health strategy or action plan (eg, prevention concordat, THRIVE, suicide prevention action plan) | ► Public health | ► MIND |
| ► Clinical Commissioning Groups (CCG) Annual Report and Accounts | ► Housing | ► Citizens Advice Bureau |
| ► Joint Health and Wellbeing Strategy | ► Community safety | ► Key Housing Associations |
| ► Annual Public Health Report | ► Community health | ► Community Grants Organisation |
| ► Joint Strategic Needs Assessment | ► Social care | ► Jobcentre Plus |
| ► The local sustainability and transformation plan | ► Adult education | ► Age UK |
| ► The Local Mental Health Strategy | ► Libraries, arts and culture | ► Age Concern |
| ► Signposting documents, websites and guides to mental health services. | ► Leisure and recreation | ► Rethink Mental Illness |
| | ► Neighbourhood and communities | ► Samaritans |
| | | ► Together |
| | | ► CALM—Campaign Against Living Miserably |
| | | ► Carer's Trust |

individuals. The list of data to extract (box 1) was adapted from the TIDieR (Template for Intervention Description and Replication) checklist[48] to be applicable for complex public health interventions. Data will be extracted from individuals either by structured interview or an electronic questionnaire with information regarding the intervention(s) of which they have knowledge.

## Data analysis

Descriptive information for each intervention will be tabulated. Interventions will be grouped according to their intervention type (examples of potential intervention types include signposting, befriending; legal or financial practical advice, education and advocacy; counselling, peer support, strategy and policy; activity/social group, green space/area regeneration), target population and intended primary outcome (eg, well-being or measures of mental distress). A descriptive comparison will be made between the selected geographical areas.

## Patient and public involvement statement

Peer researchers were recruited and coordinated by the McPin Foundation (a charity which aims to deliver more meaningful and more impactful patient and public involvement in every stage of mental health research). The peer research team was recruited to provide diverse perspectives based on their different localities, backgrounds, identities and lived experiences of mental health challenges. The role of this team is to ensure that the research conducted in the public mental health programme is in line with the lived experience of mental health and is relevant to the challenges that people face in their daily lives.

The peer researchers have been involved with the piloting of the protocol by helping to identify potentially relevant organisations and individuals in two of the selected geographical areas that we intend to map and then contacting some of these organisations and individuals to gather information about PMH interventions they are delivering. We plan to have the peer researchers involved in a similar way when mapping the remaining selected geographical areas. They will also be involved in the analysis, write up and dissemination phases of the study. They will contribute to the writing of the study report and they will advise on the most effective approach of disseminating information to community groups, voluntary sector organisations, public health practitioners and other public stakeholders and will be involved in creating the content of the dissemination. This may include producing materials such as posters and leaflets, running workshops, using social media and their hashtag #IamPublicMentalHealth, and making vlogs or writing blogs.

## ETHICS AND DISSEMINATION

Ethical approval was obtained from Durham University's Department of Sport and Exercise Sciences Research Ethics Committee (application reference: SPORT-2019-06-28T15:10:42-lxkc61). We plan to disseminate our findings at relevant conferences, meetings and through peer-reviewed journals. We will also disseminate our findings to the key personnel and organisations that participate in the study, to people who may benefit from identified interventions and the general public through social media, newsletters or local community groups. Finally, we will disseminate our findings through SPHR's PMH virtual stakeholder network which consists

**Box 1    Data to be extracted about each intervention if available (list adapted from the TIDieR (Template for Intervention Description and Replication) checklist)**

1. Name of intervention.
2. Type of intervention (befriending; practical advice, education and advocacy; counselling; peer support; strategy or policy; activity/ social group).
3. Aims and objectives (eg, what problem or risk factor does the intervention seek to address).
4. What are the intended primary and secondary outcomes?
5. Description of the content of the intervention (what activities, procedures, processes take place, schedule, duration and intensity).
6. Target population (who is eligible to take part).
7. Geographical area of delivery.
8. Setting (eg, community centre, sports club and home-based).
9. What facilities, infrastructure and support are required to deliver intervention?
10. How is it delivered (online, one-to-one and in groups)?
11. Who delivers the intervention (expertise, background and training)?
12. Who is the provider?
13. How is it funded?
14. What is the funding period?
15. Has the intervention been evaluated?
16. If the intervention has been evaluated, are the following data available:
► What were the headline conclusions?
► What was the methodology?
► What were the participants characteristics, reach and uptake?
► What was the reported impact on outcomes, was there any feedback from service users?
► Were there any unintended consequences?
► Did people adhere to the intervention?
► Were there any issues with intervention fidelity (gaps in delivery, local adaptation, modification or personalisation)?
► Is the intervention sustainable or scalable?
► Is the intervention cost-effective?

of academics, voluntary sector individuals, public health practitioners and public stakeholders.

## STRENGTHS AND LIMITATIONS OF THIS STUDY
### Strengths
A major strength of this protocol is that it uses multiple sources to identify interventions and therefore increases the likelihood of all interventions in an area being discovered, including ones that the local authority or prominent third sector organisations are not aware of themselves. Another strength is that it has already been piloted in two of the selected areas, demonstrating its feasibility, and refinements have been made (eg, we decided to only include interventions that were available during data collection period due to feasibility issues of collecting information about interventions that were no longer being delivered). It is also a strength that diverse geographical areas across are included in this study, increasing the generalisability of the mapping results.

### Limitations
A potential weakness of the protocol is that there is no guarantee that key individuals who are approached for information will be willing or able to provide the most accurate and up-to-date information about interventions and services in their area. There is also the possibility that local authority and third sector websites may not be regularly updated or that there will be variations in their comprehensiveness. However, using more than one source to obtain information about each intervention should address these limitations. Two other potential limitations are that we are focused on interventions and services that are explicitly aiming to improve mental health and that we are excluding wholly private for-profit interventions. It is possible that many services are in existence that do have a positive effect on participants' mental health but it is either not their explicit aim or they are a private enterprise, for example, a cycling group, a yoga course, a tennis club or a dance class. It should also be noted that there are other domains of well-being that are not represented in the design of this protocol. It is also a limitation that no areas in the South West of England are included in this study.

**Author affiliations**
[1]Department of Sport and Exercise Sciences, Durham University, Durham, UK
[2]Division of Psychiatry, UCL, London, UK
[3]Department of Primary Care and Population Health, UCL, London, UK
[4]Department of Primary Care and Public Health, Imperial College, London, UK
[5]Institute of Health and Society, Newcastle University, Newcastle upon Tyne, UK
[6]Institute of Public Health, University of Cambridge, Cambridge, UK
[7]McPin Foundation, London, UK
[8]Health Improvement Directorate, Public Health England, London, UK

**Acknowledgements** The authors would like to sincerely thank all of the local authority and third sector staff and volunteers who gave up their time to contribute to this study. They would also like to thank peer researchers and the McPin Foundation for their invaluable help. DO is in part supported by the National Institute for Health Research (NIHR) Collaboration for Leadership in Applied Health Research and Care (CLAHRC) North Thames at Bart's Health NHS Trust and the University College London Hospitals National Institute for Health Research Biomedical Research Centre.The NIHR School for Public Health Research is a partnership between the Universities of Sheffield; Bristol; Cambridge; Imperial; and University College London; The London School for Hygiene and Tropical Medicine (LSHTM); LiLaC—a collaboration between the Universities of Liverpool and Lancaster; and Fuse—The Centre for Translational Research in Public Health a collaboration between Newcastle, Durham, Northumbria, Sunderland and Teesside Universities

**Contributors** EJO and DO are the principal investigators. JD is the programme manager. FHD, MM, EJO, SG, DO, KRW, LL, JD, EFSK, LF, OJ, VP and JS were involved with designing the protocol methods. FHD, MM, CB, CL, OJ and other local peer researchers will be involved with applying the protocol to collect data. JK, FHD, MM, LF, VP and CB will be involved with data analysis. FHD led on the writing and editing of the manuscript. All authors contributed to the writing and editing of the protocol for publication and read and approved the final manuscript.

**Funding** This project is funded by the National Institute for Health Research (NIHR) School for Public Health Research (SPHR) (Grant Reference Number PD-SPH-2015).

**Disclaimer** The views expressed are those of the author(s) and not necessarily those of the NIHR or the Department of Health and Social Care.

**Competing interests** None declared.

**ORCID iDs**
Fiona H Duncan http://orcid.org/0000-0002-4929-5685
Kate R Walters http://orcid.org/0000-0003-2173-2430
Jude Stansfield http://orcid.org/0000-0002-7989-5630
Emily J Oliver http://orcid.org/0000-0002-1795-8448

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
