## [Reviewer comments · BMJ Open]

ARTICLE DETAILS

TITLE (PROVISIONAL)	The delivery of community-centred public mental health interventions in diverse areas in England: a mapping study protocol.
AUTHORS	Duncan, Fiona; McGrath, Mike; Baskin, Cleo; Osborn, David; Dykxhoorn, Jen; Kaner, Eileen; Gnani, Shamini; LaFortune, Louise; Lee, Caroline; Walters, Kate; Kirkbride, James; Fischer, Laura; Jones, Oli; Pinfold, Vanessa; Stansfield, J; Oliver, Emily

VERSION 1 – REVIEW

REVIEWER	Rosalie A. Torres Stone Clark University, United States of America
REVIEW RETURNED	30-Mar-2020

GENERAL COMMENTS	The researchers emphasize that mental health should be considered to be more than the absence of illness, and therefore public mental health should include mental health promotion as well as mental illness prevention. In doing so, they recognize the need to first understand what mental health approaches are being delivered in local areas. To that extent, the researchers developed a comprehensive protocol designed to identify the types of community centered interventions in selected geographic areas to improve public mental health outcomes. Will the researchers be reporting the cost effectiveness of each of these approaches in their final results?
--

REVIEWER	Sarah Stewart-Brown University of Warwick UK My research relates to public mental health, but that is presumably why you have asked me to review the paper
REVIEW RETURNED	08-Apr-2020

GENERAL COMMENTS	This paper describes the protocol for a study which aims to map public mental health interventions in carefully selected local authority areas The aims of the study are laudable. Understanding of public mental health is limited. Interventions to promote public mental health come in a great variety of guises and are offered by a wide range of providers. The paper is well written and the methodology reasonable. The use of people with lived experience of mental illness as peer researchers will however bias the study outcomes towards identifying interventions provided for this group.
---

The study design is limited by the authors' conceptualization of public mental health as illustrated by the bibliography they quote. If the study is carried out as designed it will simply serve to perpetuate this limited conceptualization and miss some of the most important public mental health interventions being provided in the community.

One key text in terms of conceptualization of Public Mental Health in the UK is the recently published Oxford Textbook of Public Mental Health Dinesh Bhugra, Kamaldeep Bhui, Samuel Yeung Shan Wong, and Stephen E. Gilman OUP 2019. This text does not seem to have been allowed to inform the study design.

Other literature which is not quoted includes the UK Faculty of Public Health report Better Mental Health for all which has published a conceptualisation of public mental health and public mental health interventions
<https://www.fph.org.uk/media/1644/better-mental-health-for-all-final-low-res.pdf>. It is odd that the publications of the Mental Health Foundation are cited but not those of the professional body overseeing public health practice in the UK

The study protocol as described should not be limited to interventions which explicitly address mental health. Until very recently the term mental health has been used to describe mental illness, and NGOs and other service providers offering wellbeing interventions have therefore intentionally avoided the label to avoid stigmatisation. Many of these NGOs are now evaluating their interventions with mental wellbeing measures like WEMWBS. Measurement of mental wellbeing could be a useful criterion to help identify local public mental health interventions.

An example of a low cost effective intervention which will be missed by the current protocol is community choirs. These demonstrably improve the mental health of people with mental and physical illness and the mental wellbeing of those who do not have a diagnosis. They would never be advertised as mental health interventions because this would cut across their ethos. Such choirs may be provided in health service and social services settings. They may receive local authority financial grants. Or they may be provided in communities at very low cost to the low or un-waged, subsidised by those who are earning or well off. This type of approach epitomises the self-help, community engagement approaches that best address mental health and wellbeing in a non-stigmatising, social capital generating way.

It is also not sufficient to limit the review to publicly funded interventions. As another example of an evidence-based wellbeing intervention that would be explicitly excluded from the study is yoga groups. This intervention may be offered in health and social services settings, but is more commonly provided in the community by private yoga instructors who usually provide subsidised teaching for the low or un-waged. The promotion of wellbeing is often an explicit aim of these groups.

The What Works for Wellbeing Centre
<https://www.google.com/search?client=firefox-b-d&q=What+Works+for+Wellbeing+Centre> has published a wide range of evidence synthesis of interventions which promote wellbeing that could greatly inform this protocol.

	The limitations section of the paper mentions some interventions which would be excluded eg yoga course, a cycling group, a dance club, but many are not mentioned. The list of other interventions which should be covered in the study should include but is not be limited to: mindfulness groups, relationships education provision, volunteering, walking groups, parenting support interventions which reliably improve parental wellbeing and school based interventions to improve teacher wellbeing.
--	---

REVIEWER	Sebastian Rosenberg Centre for Mental Health Research Australian National University Australia
REVIEW RETURNED	06-May-2020

GENERAL COMMENTS	This protocol outlines the procedure for study of an important and largely neglected area.
--

VERSION 1 – AUTHOR RESPONSE

Response to Reviewer 1

Comment requiring response	Research team response
Comment 1.1: The researchers emphasize that mental health should be considered to be more than the absence of illness, and therefore public mental health should include mental health promotion as well as mental illness prevention. In doing so, they recognize the need to first understand what mental health approaches are being delivered in local areas. To that extent, the researchers developed a comprehensive protocol designed to identify the types of community centered interventions in selected geographic areas to improve public mental health outcomes. Will the researchers be reporting the cost effectiveness of each of these approaches in their final results?	Response 1.1: Thank you for taking the time to review our article and for your positive comments. We will report the cost effectiveness of each identified intervention where this information is available and have added text to pages 6 and 11 to confirm this.

Response to Reviewer 2

Comment requiring response	Research team response
Comment 2.1: This paper describes the protocol for a study which aims to map public mental health interventions in carefully selected local authority areas. The aims of the study are laudable. Understanding of public mental health is limited. Interventions to promote public mental health come in a great variety of guises and are offered by a wide range of providers. The paper is well	Response 2.1: Thank you for your comments, and support for research in this area. We are aware that we will surface a large number and range of different projects provided by

written and the methodology reasonable. The use of people with lived experience of mental illness as peer researchers will however bias the study outcomes towards identifying interventions provided for this group.	various providers, which partly underpins the need for a mapping exercise to draw these together. We do not agree that working with peer researchers (as part of a wider team) will bias the study. Peer researchers are skilled individuals (trained and supported by the McPin Foundation, https://mcpin.org/), and the likelihood of them introducing bias is equal to anyone else on the team. Our public health peer research team do not self-identify as people with experience of 'mental illness'; they have a range of life experiences and poor mental health that mean they know their communities and organisations within them well, and how to find out information required to explore their purpose, focus and thus suitability for inclusion or exclusion in our mapping work. This team is therefore adding capacity to the project. Notwithstanding this, we used and applied generic search approaches and inclusion criteria for interventions, and data was cross-checked across the wider team, limiting opportunity for bias.
Comment 2.2: The study design is limited by the authors' conceptualization of public mental health as illustrated by the bibliography they quote. If the study is carried out as designed it will simply serve to perpetuate this limited conceptualization and miss some of the most important public mental health interventions being provided in the community. One key text in terms of conceptualization of Public Mental Health in the UK is the recently published Oxford Textbook of Public Mental Health Dinesh Bhugra, Kamaldeep Bhui, Samuel Yeung Shan Wong, and Stephen E. Gilman OUP 2019. This text does not seem to have been allowed to inform the study design. Other literature which is not quoted includes the UK Faculty of Public Health report Better Mental Health for all which has published a conceptualisation of public mental health and public mental	Response 2.2: Thank you for your comments regarding the conceptualisation of Public Mental Health. We agree that conceptualisations of PMH should allow for a wide remit (indeed, this underpins our approach to our inclusion criteria) and have made some changes to the introduction and our inclusion criteria to clarify this (see pages 4 and 7). The Better Mental Health for All report was already cited

health interventions https://www.fph.org.uk/media/1644/better-mental-health-for-all-final-low-res.pdf. It is odd that the publications of the Mental Health Foundation are cited but not those of the professional body overseeing public health practice in the UK	in our introduction (reference [27]).
Comment 2.3: The study protocol as described should not be limited to interventions which explicitly address mental health. Until very recently the term mental health has been used to describe mental illness, and NGOs and other service providers offering wellbeing interventions have therefore intentionally avoided the label to avoid stigmatisation. Many of these NGOs are now evaluating their interventions with mental wellbeing measures like WEMWBS. Measurement of mental wellbeing could be a useful criterion to help identify local public mental health interventions. An example of a low cost effective intervention which will be missed by the current protocol is community choirs. These demonstrably improve the mental health of people with mental and physical illness and the mental wellbeing of those who do not have a diagnosis. They would never be advertised as mental health interventions because this would cut across their ethos. Such choirs may be provided in health service and social services settings. They may receive local authority financial grants. Or they may be provided in communities at very low cost to the low or un-waged, subsidised by those who are earning or well off. This type of approach epitomises the self-help, community engagement approaches that best address mental health and wellbeing in a non-stigmatising, social capital generating way.	Response 2.3: Thank you for these comments. We intend our protocol to capture a wide variety of interventions which address indicators of both mental health and wellbeing. During our preliminary data collection, intervention aims such as “to reduce social isolation and loneliness”, “to increase happiness”, “to increase resilience”, “to improve emotional well-being” “to decrease stress” and “to improve confidence and/or self-esteem” were interpreted as a way of expressing an aim to improve mental health and/or well-being and these interventions were included. In addition, recognising that organisations sometimes intentionally avoid reference to mental health as you suggest, our protocol allows for the inclusion of interventions that do not reference mental health explicitly via public-facing documents, if intervention providers indicate this was a programme aim. Importantly, it was not the intention of this exercise to map all interventions that could have an impact on mental health and wellbeing (replicating existing work and beyond the capacity of the project), but rather to focus in depth on those that are being designed and delivered to have an impact on mental health and wellbeing. We agree that this was unclear in the protocol and have now

	clarified our aims and inclusion criteria to reflect this (see pages 6 and 7).
Comment 2.4: It is also not sufficient to limit the review to publicly funded interventions. As another example of an evidence-based wellbeing intervention that would be explicitly excluded from the study is yoga groups. This intervention may be offered in health and social services settings, but is more commonly provided in the community by private yoga instructors who usually provide subsidised teaching for the low or un-waged. The promotion of wellbeing is often an explicit aim of these groups.	Response 2.4: We appreciate that this protocol will result in wholly private for profit interventions being excluded. We adopted this approach for two reasons: first, these are problematic to identify and delimit – any area may contain hundreds of small businesses aiming to support wellbeing; second, stakeholder consultations conducted as part of the wider programme of this research directed us towards a focus on publicly and charitably funded services. We appreciate that narrowing the scope of the protocol in this way is a limitation (and have acknowledged this on page 14), but by doing so, it enables there to be more focus on innovative local authority and third sector initiatives.
Comment 2.5: The What Works for Wellbeing Centre https://www.google.com/search?client=firefox-b-d&q=What+Works+for+Wellbeing+Centre has published a wide range of evidence synthesis of interventions which promote wellbeing that could greatly inform this protocol.	Response 2.5: We agree, and did not want to replicate their work (see Response 2.3 above). Many of the domains that promote well-being highlighted by the WWCFWB have been included in our protocol. We appreciate that, for feasibility reasons, not all the domains are represented in the design of our protocol and that this is a limitation (acknowledged on page 14).
Comment 2.6: - The limitations section of the paper mentions some interventions which would be excluded e.g. yoga course, a cycling group, a dance club, but many are not mentioned. The list of other interventions which should be covered in the study	Response 2.6: We completely agree and our protocol does not exclude these IF they are intentionally aiming to

should include but is not be limited to: mindfulness groups, relationships education provision, volunteering, walking groups, parenting support interventions which reliably improve parental wellbeing and school based interventions to improve teacher wellbeing.	increase mental health and/or wellbeing. As mentioned above, preliminary results have identified many of these other interventions you have mentioned such as walking groups, various activity and social groups, maternal mental health (midwives trained to look out for mental health issues in new mothers and fathers), community cafes, befriending services, mindfulness classes, advocacy for carers, home help services, adult learning, arts and creativity groups, a social inclusion football league, community living rooms, stress workshops, workplace well-being courses and more. A number of the interventions that we have already identified have used the WEMWBS to evaluate well-being, although many have not. Although we do appreciate that some interventions will be missed, the protocol will still lead to the identification of a wide range of interesting and useful interventions.
---	--

Response to Reviewer 3

Comment requiring response	Research team response
Comment 3.1: This protocol outlines the procedure for study of an important and largely neglected area.	Response 3.1: Thank you for taking the time to review our article and for your positive comments.